# Effects of dopamine on reinforcement learning and consolidation in Parkinson's disease

**John P Grogan[1]\*[†], Demitra Tsivos[2], Laura Smith[1], Brogan E Knight[2], Rafal Bogacz[3], Alan Whone[1], Elizabeth J Coulthard[1,2]\***

[1]Institute of Clinical Neurosciences, School of Clinical Sciences, University of Bristol, Bristol, United Kingdom; [2]Clinical Neurosciences, North Bristol NHS Trust, Bristol, United Kingdom; [3]MRC Brain Network Dynamics Unit, Nuffield Department of Clinical Neurosciences, University of Oxford, Oxford, United Kingdom

**Abstract** Emerging evidence suggests that dopamine may modulate learning and memory with important implications for understanding the neurobiology of memory and future therapeutic targeting. An influential hypothesis posits that dopamine biases reinforcement learning. More recent data also suggest an influence during both consolidation and retrieval. Eighteen Parkinson's disease patients learned through feedback ON or OFF medication, with memory tested 24 hr later ON or OFF medication (4 conditions, within-subjects design with matched healthy control group). Patients OFF medication during learning decreased in memory accuracy over the following 24 hr. In contrast to previous studies, however, dopaminergic medication during learning and testing did not affect expression of positive or negative reinforcement. Two further experiments were run without the 24 hr delay, but they too failed to reproduce effects of dopaminergic medication on reinforcement learning. While supportive of a dopaminergic role in consolidation, this study failed to replicate previous findings on reinforcement learning.

**\*For correspondence:** john.grogan@bristol.ac.uk (JPG); elizabeth.coulthard@bristol.ac.uk (EJC)

**Present address:** [†]School of Clinical Sciences, University of Bristol, Bristol, UK

**Competing interests:** The authors declare that no competing interests exist.

## Introduction

Phasic changes in dopamine level are believed to encode the reward prediction error (RPE), which measures the difference between the reward expected after an action, and the reward actually received (*Schultz et al., 1993*, *Schultz et al., 1997*). In turn the RPE guides reinforcement learning (RL) such that behaviour is adapted to changing surroundings. Several studies have taken advantage of the dopaminergic depletion in Parkinson's disease (PD) in the substantia nigra pars compacta and ventral tegmental area (*Agid et al., 1989*; *Shulman et al., 2011*). PD patients are frequently treated with dopamine replacement therapy (levodopa and dopamine agonists), and thus by comparing patients in ON and OFF medication states the effects of dopamine depletion can be investigated.

One influential study using such a procedure found that dopaminergic state modulated RL from positive and negative feedback (*Frank et al., 2004*). This study used the Probabilistic Selection Task (PST), in which participants see two Japanese Hiragana symbols on the screen at the same time, and must pick one, receiving either 'Correct' or 'Incorrect' feedback (see *Figure 1*). This feedback is determined probabilistically, so that card A in pair AB is given positive feedback on 80% of trials, and negative feedback on 20%, and vice versa for card B. Pairs CD and EF have probabilities 70–30% and 60–40%, respectively. During the learning trials, if a participant chooses card A over card B, this could be because they have learned that card A is more often rewarded and should be chosen, or that card B is more often punished and should be avoided – one cannot tell these apart from this choice. So, a novel pairs test is given where all the cards are shown in all possible combinations (e.g.

**eLife digest** Brain cells release a naturally occurring chemical called dopamine. The release of this chemical affects how people respond to their ever-changing environment, including how they learn from rewards and punishments. Parkinson's disease is a condition where the brain cells that make dopamine start to die, and so the levels of dopamine in the brain begin to drop. Parkinson's disease patients are routinely given drugs to bring their dopamine levels back up to near-normal levels.

About 13 years ago, researchers found that when patients with Parkinson's disease were given dopamine-medication they were better at learning from rewards and worse at learning from punishments. If the patients were withdrawn from their dopamine-medications they were worse at learning from rewards but better at learning from punishments. However, it was not clear if this was because the dopamine affects the learning process, or if it affects how people remember what they learned and how they make choices later on.

To better understand how dopamine is involved in learning in people with Parkinson's disease, Grogan et al. looked at the effects of dopamine on memory over a timescale of 24 hours. People with Parkinson's disease and healthy volunteers were shown a choice of symbols and given the chance to learn which gave a reward – a picture of a smiling face – and which gave a punishment – a frowning face. If the Parkinson's disease patients had taken their dopamine-medication before learning the task, their memory did not worsen over the next 24 hours. This suggests that having dopamine in the brain around the time of learning helped the patients to store the memory.

The patients, however, were not any better at learning from rewards when taking their medication, which contradicts some earlier studies. To explore this further, Grogan et al. copied the exact same task from the 13-year-old study, and still did not find that patients were better at learning from reward when taking dopamine.

These findings could help scientists to better understand what dopamine does during learning and memory, and how the brain normally works. Finally, Parkinson's disease causes problems with memory. A clearer picture of the types of memory problems patients have, and of how their dopamine-medication can help, might make it easier for clinicians to treat patients with Parkinson's disease.

AB, AC, AD, AE...), without feedback, and from this the percentage of times that card A is chosen, and card B avoided, can be used as benchmarks for positive and negative reinforcement, respectively.

PD patients chose A more and avoided B less when ON medication, and vice versa for OFF medication (*Frank et al., 2004*). This suggested better learning from positive reinforcement and poorer negative reinforcement ON medication, while patients OFF showed the opposite pattern. Importantly, patients OFF medication were better at negative reinforcement than healthy age-matched controls (HC), suggesting that PD actually improved some aspect of RL. The explanation for these effects was provided with a model of the basal ganglia. In the Go-NoGo model, the two main pathways from the striatum are proposed to underlie positive and negative reinforcement. The direct pathway, which is mainly activated by striatal neurons containing dopamine D1 receptors and therefore is activated by a dopamine increase during a positive RPE, underlies positive reinforcement. The indirect pathway which is inhibited by D2 receptors, and therefore activated when a dopamine decrease signals a negative RPE, allows negative reinforcement. When PD patients are ON medication, the higher dopamine levels activate D1 receptors and inhibit D2 receptors, thus biasing towards the direct pathway, improving positive reinforcement. When OFF medication, the lower dopamine levels mean less D1 activation, and less D2 inhibition, thus increasing indirect pathway activity, and improving negative reinforcement.

While this view is persuasive, some more recent studies have cast doubts on the extent to which dopamine is involved in the learning part of RL, and how much is the expression of that learning. In *Frank et al. (2004)* study, the RL test was given immediately after learning, and it was only in this test that the effects were seen. However, as patients were ON for both learning and testing, or OFF

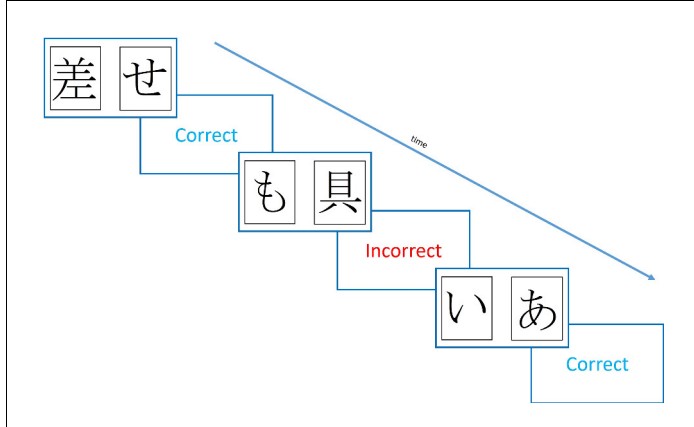

**Figure 1.** Diagram of the learning trials of the Probabilistic Selection Task. In each pair one **card** is more likely to be rewarded (shown 'Correct' feedback) than the other, with card A in pair AB rewarded 80% of trials, and card B on 20% of trials. For pairs CD and EF, the probabilities are 70–30% and 60–40%.

for both, it meant that the dopaminergic effects could have occurred at either point. One study attempted to correct for this by adding a one hour delay between learning and testing, allowing PD patients to learn OFF medication, then be tested ON medication (along with ON-ON and OFF-OFF conditions) (*Shiner et al., 2012*). It was found that being ON medication at the time of testing produced greater expression of positive reinforcement than being OFF, while it had no effect during learning. Additionally, RPE signals in the striatum during learning were not affected by medication state, but ventromedial PFC and nucleus accumbens signals in the test phase only tracked the value of the stimuli when patients were ON medication. This suggests that dopamine may not play as much of a role in learning, but instead influence the choices made after the expected rewards have been learnt. Other studies have also shown dopaminergic D2 receptor effects mainly on the expression rather than learning (*Eisenegger et al., 2014*; *Pessiglione et al., 2006*).

The direct link between dopamine and RL, as opposed to expression of information, was further questioned by another study, which found that dopaminergic effects could be shown even when rewards were not actually given during learning (*Smittenaar et al., 2012*). Participants received one of two shapes, probabilistically, after selecting a stimulus, and only after the learning trials were they told that one shape corresponded to winning money, and one to losing money. Thus, the reward/monetary associations were created separately to the stimulus-outcome associations. However, they still found that PD patients ON medication during testing (after finding out about the money), showed higher accuracy on the most rewarded stimulus, and lower accuracy on the most punished stimulus. This shows that it is possible to generate this effect without any reinforcement learning actually taking place, suggesting dopamine influences value-based decision making.

A recent extension to standard RL models offers a mechanism by which dopamine may influence expression of learning. The OpAL model (*Collins and Frank, 2014*) has separate learning rates and choice parameters for the direct and indirect pathways, which learn from positive and negative reinforcement, respectively. By allowing dopamine to affect the choice parameter, it can bias towards choosing the stimulus that learned mainly from the direct pathway, or from the indirect pathway, thus lending more weight to the positive or negative reinforcement the stimulus received.

In addition to evidence of dopamine affecting expression of learned values, there is also evidence of it affecting consolidation. PD patients ON medication showed an increase in accuracy on an RL task after a 20 min delay, while those OFF medication showed a large decrease (*Coulthard et al., 2012*). This was despite all PD patients showing the same behaviour during the learning trials. It is still possible that this is a retrieval effect, and that it was simply not seen during the learning trials as the values were still being updated, but it is also possible that the dopaminergic medication preserved the synaptic weight changes induced during learning, thus improving memory for the learned items.

This explanation ties in nicely with models of synaptic consolidation based on the synaptic tagging and capture hypothesis (*Clopath et al., 2008*; *Frey and Morris, 1998*; *Redondo and Morris, 2011*). In these models, early synaptic changes are induced during learning, but decay away unless actively prolonged, which is achieved by the changes setting a 'tag' which plasticity-related proteins (PRPs) must act on to make the changes permanent. Dopamine is hypothesised to set the threshold for the synthesis of these PRPs, so that higher levels of dopamine mean a lower threshold. Thus, PD patients OFF medication would have a higher PRP threshold, and therefore lower consolidation of early synaptic changes, leading to a delayed impairment of memory, as seen in *Coulthard et al. (2012)*.

Here, we sought to investigate the hypotheses that:

- Dopamine during learning improves delayed memory for information learned through reinforcement;
- Dopamine during learning affects choice performance in a novel pairs task 24 hr later and;
- Dopamine during testing (24 hr after learning) affects choice performance.

These hypotheses were tested in experiment 1 where surprisingly we did not show the expected effects of dopamine on novel pairs choices. We undertook a further two experiments to investigate the effects of dopamine and delays on RL, and the effects of procedural changes to the PST.

## Results

### Experiment 1

#### Learning

Eighteen PD patients and 18 HC learned a modified version of the PST (2 pairs, different probabilities of reward, smiling or frowning faces as feedback, see Figure 7), and were tested immediately, 30 min and 24 hr later. PD patients were ON or OFF their dopaminergic medication on day 1 during learning, and ON or OFF on day 2 for testing (4 conditions; within-subject design). No data were excluded, but the final learning blocks were missing from two conditions for the same participant due to experimenter and computer error.

Participants were able to learn the task, with final mean accuracies of 78.60% (SEM = 3.84) ON medication, 77.71% (4.16) OFF medication and 79.03% (4.67) for HC. No effects of medication (p=0.882, $\eta_p^2$ =0.001), or disease (p=0.846, $\eta_p^2$ =0.0004) were seen on the learning trials. We also examined win-stay lose-shift behaviour on the learning trials, but found no significant differences between groups (p>0.2; see Appendix 1 for details).

We fit variations of the Q-learning model to the participants' data, with and without different learning rates for the different PD medication states during learning (see Appendix 2 for details). A dual learning rate Q-learning model where the learning rates did not differ by medication state provided the best fit, further suggesting that dopaminergic medication did not affect learning.

#### Memory

The memory tests presented the same pairs as the learning trials, but without any feedback. The 30 min memory test data were missing from two conditions for the same patient due to experimenter and computer error.

We looked at the difference between immediate and 30 min delayed memory blocks, and between 30 min and 24 hr blocks. *Figure 2* shows that both day 1 OFF conditions (and HC) have mean decreases in memory scores, while both day 1 ON conditions have slight increases in mean memory scores, although the standard error bars overlap with zero for the latter. A two-way repeated-measures ANOVA (day 1 medications * day 2 medication) showed no significant effect of day 1 medication state on the difference between 24 hr and 30 min memory scores (F (1, 16) = 2.803, p=0.114, $\eta_p^2$ =0.149), and no other effects were significant (p>0.3, $\eta_p^2$ <0.064). T-tests comparing ON-ON and OFF-ON showed no effect for either measure (p=0.51, *d* = 0.2314; p=0.376, *d* = 0.3483). Comparing ON-OFF and OFF-OFF showed no difference in the change across 30 min (p=0.292, *d* = 0.3233), but did show significant difference in the change across 24 hr (t (16)=2.894, p=0.0106, *d* = 0.4959). This survived Bonferroni correction for four comparisons (α = 0.0125). Non-parametric Wilcoxon's tests demonstrated the same results (ON-OFF vs OFF-OFF: p=0.011, all

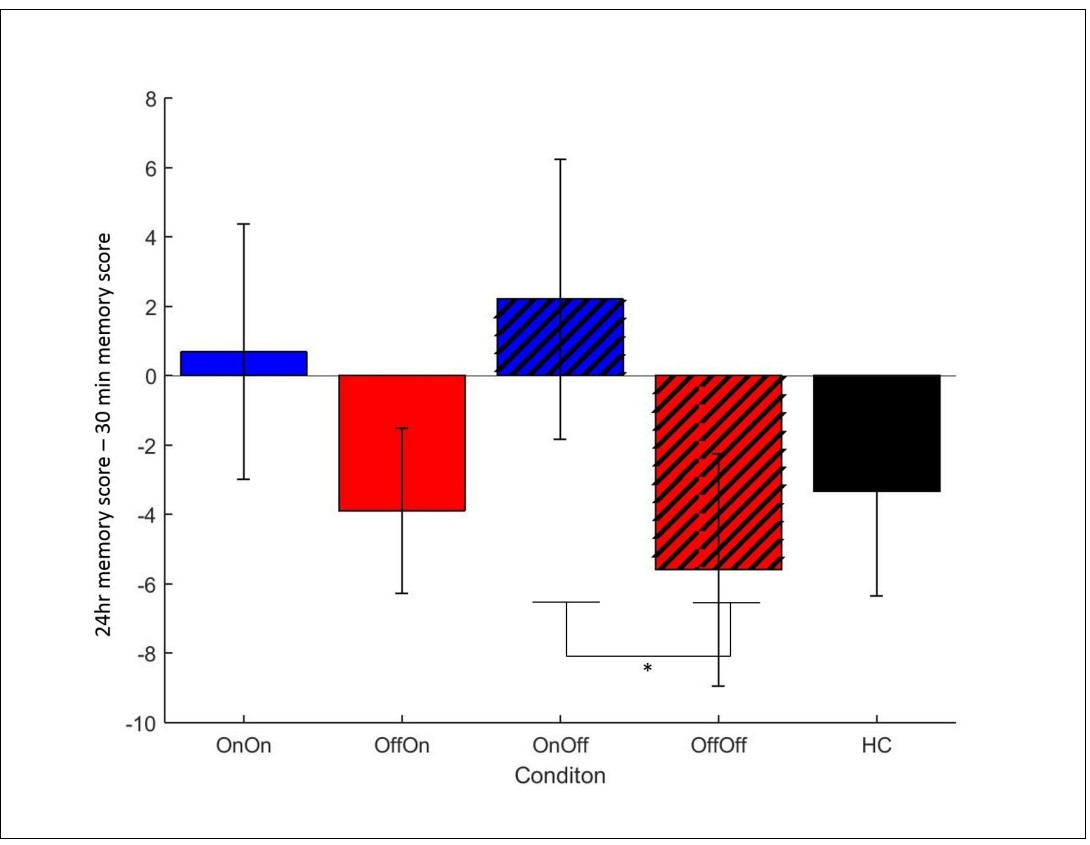

**Figure 2.** Mean difference in memory block accuracy between 24 hr and 30 min, for each condition in experiment 1. ON-OFF had significantly greater increases in memory score over this time than OFF-OFF (* = p=0.0106) and indeed both day 1 ON conditions (blue bars) had a mean increase in accuracy while both day 1 OFF conditions (red bars) and HC (black bars) had a decrease. Error bars are SEM. *Figure 2—source data 1* shows the summary statistics.

The following source data is available for figure 2:

**Source data 1.** Summary statistics for *Figure 2*, the difference in memory scores between 30 min and 24 hr tests for each condition.

others p>0.1). This suggests that low dopamine levels during learning or early memory may affect the persistence of the reinforced memory, at least in certain circumstances.

### Novel pairs

The novel pairs task was given 24 hr after learning on day 2. Each possible combination of the cards was shown six times, without feedback, in a random order. The percentage of times participants chose the most rewarded card (A) and avoided the least rewarded card (B) were used as measures of expression of positive and negative reinforcement, respectively. One block of novel pairs data were missing due to computer error.

The overall accuracy on the novel pairs test showed a positive correlation with the accuracy in the final learning block (r = 0.4365, p<0.0001), which is not surprising as participants who learned the task well would be expected to perform better on the test.

A between-subjects multivariate ANOVA was run to compare PD patients against HC (*Figure 3*). Choose-A and avoid-B were not significantly different between PD patients and HC (p=0.714, *d* = 0.0969; p=0.753, *d* = 0.0834).

We found no effects of dopaminergic state during day 1 or day 2 on expression of positive and negative reinforcement. A repeated measures ANOVA (choice * day 1 * day 2) on the PD patients'

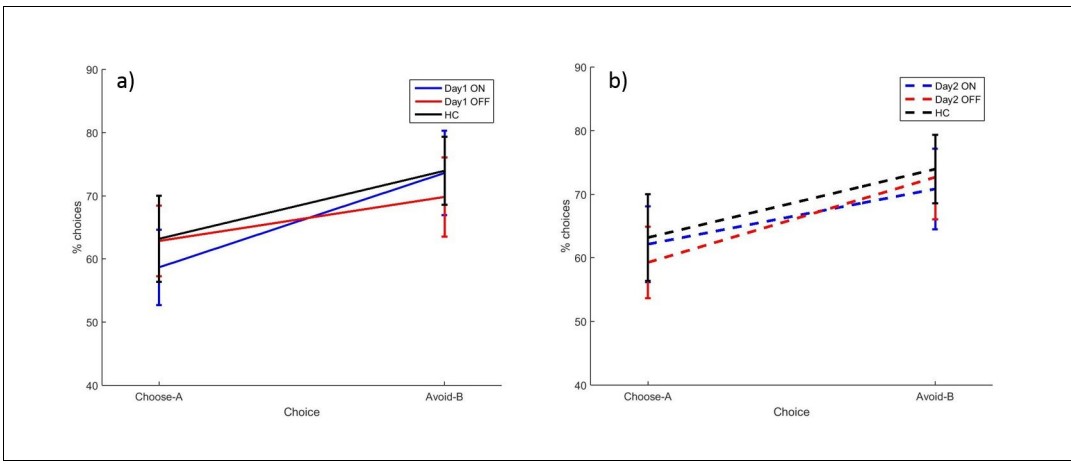

**Figure 3.** The mean percentages of choose-A and avoid-B selections at the 24 hr novel pairs tests in experiment 1, split by a) day 1 and b) day 2 conditions. There were no significant effects of day 1 or day 2 medication state (p>0.28). Error bars are SEM. *Figure 3—figure supplement 1* shows the data when filtered by performance on the 80–20% pair for day 1 conditions, and *Figure 3—figure supplement 2* shows the filtered day 2 conditions. *Figure 3—source data 1* shows the summary statistics.

The following source data and figure supplements are available for figure 3:

**Source data 1.** Summary statistics for *Figure 3*, the percentages of choose-A and avoid-B selections in the experiment 1 novel pairs test.

**Figure supplement 1.** The mean percentages of choose-A and avoid-B selections for the filtered data in the experiment 1 novel pairs test split by day 1 conditions.

**Figure supplement 1—source data 1.** Summary statistics for *Figure 3—figure supplement 1*, percentages of choose-A and avoid-B behaviours for experiment 1 split by Day 1 condition, after data filtering.

**Figure supplement 2.** The mean percentages of choose-A and avoid-B selections for the filtered data in the experiment 1 novel pairs test split by day 2 conditions.

**Figure supplement 2—source data 1.** Summary statistics for *Figure 3—figure supplement 2*, percentages of choose-A and avoid-B behaviours for experiment 1 split by day 2 condition, after data filtering.

data showed a significant effect of choice (F (1, 16)=4.692, p=0.046, $\eta_p^2$ = 0.227), with avoid-B higher than choose-A overall. There were no significant effects of day 1 or day 2 medication state, or any interactions (p>0.12). This suggests that overall PD patients were better at expressing negative reinforcement, but that medication on day 1 or day 2 had no effect.

There were also no significant effects when examining individual conditions with paired t-tests for choose-A (p>0.08, $\alpha$ = 0.0125) or avoid-B selections (p>0.2). This suggests that neither day 1 nor day 2 medication state affected choose-A or avoid-B performance.

The data filtering used by *Frank et al. (2004)* was applied to the data here also. Any participants who failed the easiest choice (A vs B) on 50% or more of the novel pairs trials were assumed to not have learned the task properly, and that condition's novel pairs data were excluded. The other conditions from the same participant could still be included if they passed the filtering. This lead to 4/18 ON-ON blocks excluded, 2/17 OFF-OFF blocks, and 3/18 for all other conditions. No significant effects of choice or medication state were seen on the filtered data (p>0.1; *Figure 3—figure supplements 1* and *2*).

Overall, there were no significant effects of medication or disease state, suggesting that PD and dopaminergic medication did not affect novel pairs performance on the modified PST when tested 24 hr later. As experiment 1 failed to show the expected effects of dopamine on RL, experiment 2

was run without the 24 hr delay to attempt to replicate effects of dopamine on RL when tested immediately after learning.

## Experiment 2

### Learning

Eighteen PD patients and 20 HC completed the modified PST with the novel pairs test immediately after learning. There were no memory blocks. PD patients were ON or OFF medications for both learning and testing. Participants again reached over 70% accuracy on the learning trials on average (ON: 77.01% (SEM = 3.67), OFF: 78.06% (4.04), HC: 74.69% (4.43)), and there were no effects of medication (p=0.806, $\eta_p^2 = 0.004$) or disease (p=0.563, $\eta_p^2 = 0.006$).

### Novel pairs

Overall accuracy in the novel pairs tests correlated positively with accuracy in the final learning block (r = 0.6686, p<0.0001). A multivariate ANOVA comparing PD patients and HC showed no effects of disease state on choose-A (p=0.285, $\eta_p^2 = 0.021$) or avoid-B (p=0.226, $\eta_p^2 = 0.027$). Paired samples t-tests to compare ON and OFF medication conditions showed no significant differences for choose-A (p=0.727, $d = 0.1311$) or avoid-B (p=0.580, $d = 0.1910$; *Figure 4a*). Filtered data were also analysed; 1/18 ON, 3/18 OFF and 3/20 HC blocks were filtered out, and no effects of condition were

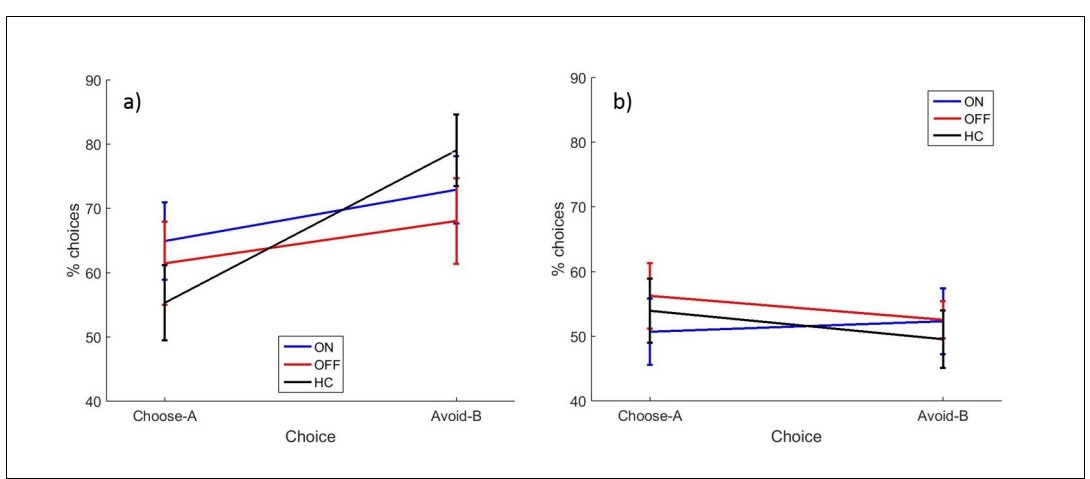

**Figure 4.** The mean percentages of selections on the novel pairs test for a) experiment 2 and b) experiment 3. There were no significant effects of disease state or medication condition on either selection in either experiment 2 (p>0.2) or experiment 3 (p>0.3). Error bars are SEM. *Figure 4—figure supplement 1* shows the data after filtering was applied to experiment 2, and *Figure 4—figure supplement 2* shows the filtered data for experiment 3. *Figure 4—source data 1* shows the summary statistics.

The following source data and figure supplements are available for figure 4:

**Source data 1.** Summary statistics for *Figure 4*, the percentages of choose-A and avoid-B selections in experiments 2 and 3 novel pairs tests.

**Figure supplement 1.** The mean percentage of selections for the filtered data for experiment 2.

**Figure supplement 1—source data 1.** Summary statistics for *Figure 4—figure supplement 1*, percentages of choose-A and avoid-B behaviours for experiment 3, after data filtering.

**Figure supplement 2.** The mean percentage of selections for the filtered data for experiment 3.

**Figure supplement 2—source data 1.** Summary statistics for *Figure 4—figure supplement 2*, percentages of choose-A and avoid-B behaviours for experiment 2, after data filtering.

found (p>0.1; *Figure 4—figure supplement 1*). Thus we did not find evidence that dopaminergic medication state affects choices on the modified-PST as in previous studies (*Frank et al., 2004*).

## Experiment 3

As experiment 2 did not replicate the findings of *Frank et al. (2004)*, an exact replication was run to ensure we observed the well-described effect of dopamine enhancing positive reinforcement or impairing negative reinforcement.

### Learning

Eighteen PD patients and 18 HC completed the original PST (*Frank et al., 2004*). PD patients were ON or OFF for both learning and novel pairs test. Neither PD nor medication state significantly affected the number of blocks completed (ON: 4.78 (0.59), OFF: 5.28 (0.50), HC: 5.94 (0.42); PD: p=0.145, $\eta_p^2$ = 0.040; medications: p=0.477, $\eta_p^2$ = 0.030) or the final learning accuracy (ON: 61.94% (3.35), OFF: 61.67% (3.26), HC: 57.22% (3.97); PD: p=0.291, $\eta_p^2$ = 0.021; medications: p=0.950, $\eta_p^2$ = 0.0002). Note that the final learning accuracies were lower than in experiments 1 and 2.

### Novel pairs

Again, overall accuracy in the novel pairs test correlated positively with the accuracy in the final learning block (r = 0.4680, p=0.0004). PD patients did not differ from HC on either choice (p=0.940, $\eta_p^2$ = 0.0001; p=0.577, $\eta_p^2$ = 0.006). Paired-samples t-tests showed no significant differences in choose-A (p=0.363, *d* = 0.2573) or avoid-B selections for the two medication conditions (p=0.968, *d* = 0.0132), meaning dopaminergic medication state didn't affect positive or negative reinforcement in PD patients (*Figure 4b*). Data filtering excluded 8/18 ON blocks, 6/18 OFF blocks and 7/18 HC blocks, and no significant effects of disease or medication were seen in the remaining data (p>0.5; *Figure 4—figure supplement 2*).

It should be noted that participants were only just above chance performance on the novel pairs task in experiment 3, with mean percentage of choices between 49% and 57% for all groups. This is lower than in experiments 1 and 2, suggesting that the original PST was harder for participants to learn.

## General results

### Learning

As each experiment provided related statistics of learning, they were analysed together to get the full picture. A univariate analysis was run on the final learning block accuracies for each experiment (*Figure 5*), which showed a significant effect of experiment (F (2, 187)=18.416, p<0.000001, $\eta_p^2$ =0.165), but no effects of condition (p=0.578, $\eta_p^2$=0.015) or interaction (p=0.587, $\eta_p^2$ = 0.015). Post-hoc comparisons with Bonferroni corrections showed that experiment 3 had significantly lower final learning block accuracies than experiments 1 (p<0.000001) and 2 (p=0.000002), but there were no differences between experiments 1 and 2 (p=1).

### Novel pairs

A multivariate ANOVA was run on choose-A and avoid-B from each experiment. ON and OFF in experiments 2 and 3 were treated the same as ON-ON and OFF-OFF from experiment 1. There were no significant effects of condition on either

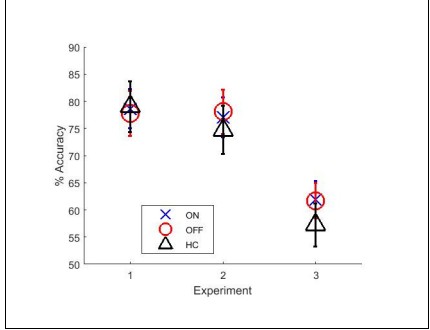

**Figure 5.** Mean final learning block accuracies across the three experiments. Experiment 3's final accuracy was significantly lower than experiments 1 and 2's (p<0.000001). Error bars are SEM. *Figure 5—source data 1* shows the summary statistics.

The following source data is available for figure 5:

**Source data 1.** Summary statistics for *Figure 5*, the mean final learning block accuracies across the three experiments.

choice (p=0.609, $\eta_p^2$ = 0.014; p=0.583, $\eta_p^2$ = 0.015), but were significant effects of experiment on avoid-B (F (2, 188)=16.069, p<0.000001, $\eta_p^2$ = 0.146). Bonferroni-corrected post-hoc tests showed that experiment 3 had lower avoid-B than experiments 1 (p=0.000005) and 2 (p=0.000013), while experiments 1 and 2 did not differ (p=1). There was no effect of experiment on choose-A performance (p=0.327, $\eta_p^2$ = 0.012), and no interaction of experiment and condition for either choice (p=0.556, $\eta_p^2$ = 0.016; p=0.749, $\eta_p^2$ = 0.010).

This means that experiment 3 had poorer final learning accuracy (despite passing the accuracy thresholds) and less avoid-B choices on the novel pairs test, than experiments 1 and 2 which used the modified-PST. Interestingly, this was because experiments 1 and 2 had avoid-B scores higher than choose-A, while experiment 3 had about the same scores.

## Discussion

These experiments found that dopaminergic medication during learning prevented a decrease in memory for RL over 24 hr. However, dopamine did not affect expression of positive or negative reinforcement either during learning or testing, when tested immediately or 24 hr after learning. PD patients did not differ from HC in any of the experiments. Finally, experiment 3, using the original PST, had much lower accuracy in the learning trials, and lower avoid-B scores than experiments 1 and 2 which used the modified-PST.

### Effects of dopamine on consolidation

Dopaminergic medication seemed to affect memory performance on the PST. When patients were OFF medication on day 2, being ON medication the day before (during learning) prevented a decrease in their memory over the 24 delay. Interestingly, both day 1 ON conditions had a pattern of a slight increase in memory scores (albeit non-significantly different to zero), while both day 1 OFF conditions (and HC) showed a decrease on average. As this score was the difference between 30 min and 24 hr delay tests, it suggests that day 1 dopamine improved consolidation of the learned values sometime after 30 min, preventing a decay in the memory. This is in line with a previous study showing a benefit of dopamine at the time of learning on memory testing 20 min later (*Coulthard et al., 2012*), although it was only seen at longer delays here.

All PD patients went back ON medication immediately after the day 1 session regardless of day 2 condition, so all patients were in an ON state for the hours after learning (see *Figure 6* for diagram). This means the day 1 ON and OFF groups differed in dopaminergic activity until about 1.5 hr after learning, when the medication would have reached peak concentration. This gives a short time window for day 1 medication to affect consolidation of RL.

This finding fits with the wider literature implicating dopamine in memory and consolidation mechanisms (*Lisman et al., 2011*; *Shohamy and Adcock, 2010*). Dopamine given before or after learning can improve consolidation (*Bernabeu et al., 1997*; *de Lima et al., 2011*; *Furini et al., 2014*; *Péczely et al., 2016*; *Rossato et al., 2009*), although there is still debate on the time course of its effects. Synaptic tagging and capture models suggest dopaminergic effects would take place within a few hours of learning (*Clopath et al., 2008*; *Redondo and Morris, 2011*), and consolidation effects on RL have been reported over shorter times before (*Coulthard et al., 2012*). Synaptic tagging has mainly been studied in hippocampal circuits, and may relate to the binding of separate experiences within a time window of hours or days (*Shohamy and Adcock, 2010*). However, the PST is assumed to rely on basal ganglia functioning, at least when there are short delays between action and feedback as there were here (*Foerde and Shohamy, 2011*). Combining computational synaptic tagging and capture models with basal ganglia RL models would show whether such an explanation could explain this effect. Further behavioural work fractioning the time window after learning where dopamine impacts on consolidation of RL could also be illuminating.

Consolidation has not often been the focus in RL studies, rather learning or testing effects, but a few studies have shown that RL consolidation is affected by dopamine or sleep. Three studies have found that sleep affects performance on the Weather Prediction Task (*Barsky et al., 2015*; *Djonlagic et al., 2009*; *Lerner et al., 2016*). While this task is different to the PST, it is not unreasonable to expect sleep to also affect other RL tasks similarly. Dopamine has also been shown to affect sleep consolidation for reward-related memory (*Feld et al., 2014*), and while reward-related

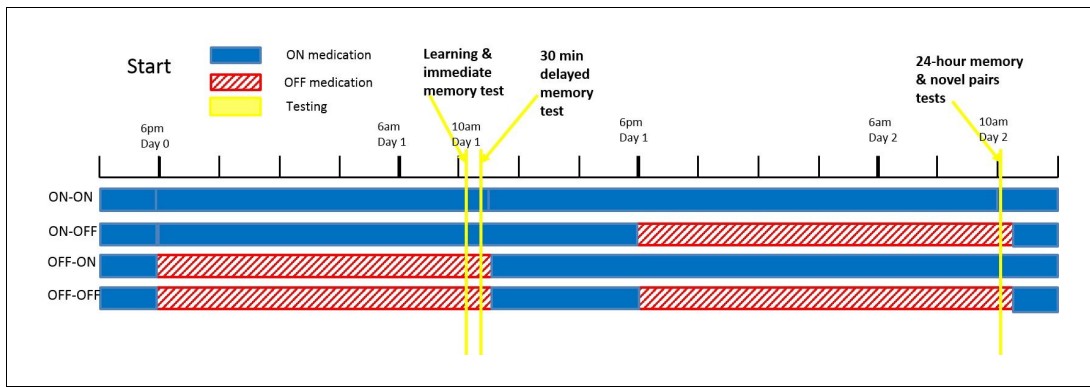

**Figure 6.** A diagram of the timing of PD medication withdrawal for all four conditions in experiment 1. Blue is when patients were ON medication, red hatched bars when they were OFF, and yellow bars the PST phases. In order for patients to be fully OFF medication during testing, they were withdrawn from their dopaminergic medications a minimum of 15 hr prior to testing (>24 hr for long-lasting medications). Note that in all conditions, patients were ON medication for a few hours after the day 1 session, to minimise the time spent OFF medication. DOI: 10.7554/eLife.26801.020

memory is partly a declarative memory process, it relies on the same reward-processing brain regions that underlie RL (*Wittmann et al., 2005*).

An alternate explanation is that dopamine state during the 30 min memory block affected reconsolidation of the RL values, allowing the values to be reconsolidated properly and recalled accurately the next day. Dopamine has been implicated in reconsolidation (*Rossato et al., 2015*), although it has not been investigated in the RL domain.

## Effects of dopamine on learning from positive and negative feedback

Experiment 1 sought to separate the effects of dopamine during learning and during testing on positively and negatively reinforced information, but found no effects of either. PD patients also did not differ from HC. If our study were simply underpowered we might expect the results to at least be in the direction predicted by previous studies. Interestingly, the direction of effect was opposite to the expected effect, with the day 1 ON conditions having the highest amount of avoid-B selections. The classic view is that dopamine improves positive reinforcement, at the cost of impaired negative reinforcement, so it is unclear why the condition in which patients have the most dopamine would show greatest expression of negative reinforcement.

Due to this unexpected pattern, another experiment was run without the 24 hr delay between learning and testing, to try to replicate the expected pattern of behaviour. Experiment 2 used the same modified PST as experiment 1, but with testing immediately after learning. Again, PD patients did not differ from HC, and showed no effect of medication. This is in contradiction to previous studies which found that PD patients had greater expression of positive reinforcement when ON medication, and greater expression of negative reinforcement when OFF medication (*Frank et al., 2004*; *Shiner et al., 2012*).

It is surprising that we were unable to replicate the findings of dopamine affecting positive and negative reinforcement, especially in experiment 3 which was designed to be an exact replication of the original study (*Frank et al., 2004*). We now look at the possible differences between the studies. The main results reported here were on unfiltered data, but when the data filtering used in previous studies was applied, it made little difference to the results.

The average accuracy on novel pairs in experiment 3 was much lower than reported in *Frank et al. (2004)*, where the patients ON medications achieved 78% accuracy in choosing A and patients OFF medications achieved 82% accuracy in avoiding B. By contrast the corresponding accuracies in our experiment were very close to chance (53% and 51%, respectively; 54.5% and 46.7% for filtered data). So, since the patients were unable to well express any learned preferences, it is not surprising that there was no difference in expression preference learned from positive and negative feedback.

One important thing to consider is whether there were sample differences that could explain the disparity between our results and previous studies (e.g. *Frank et al., 2004*). Our samples were very closely matched in age, gender and disease severity to the PD patients tested ON medication in *Frank et al. (2004)*. T-tests on the *Frank et al. (2004)* PD sample and our experiment 3 sample showed no significant differences in ages or Hoehn and Yahr stages, and $\chi^2$ tests showed no significant difference in gender distributions between PD and HC (p>0.1). The PD patients (p<0.001) and HC (p=0.046) in experiment 3 had fewer years of education on average than those in *Frank et al. (2004)*, which could have contributed to the failure to learn this task, although our experiment 1 sample also had fewer years of education (p<0.001) and they could learn the modified PST. Given that other studies have found effects of dopamine on RL tasks when using different samples, the differences here are unlikely to be dependent on demographic characteristics.

Many studies have used the PST (or variations), and while the findings are not always exactly the same, the general pattern of higher dopamine levels either improving expression of positive reinforcement (*Rutledge et al., 2009*; *Shiner et al., 2012*; *Smittenaar et al., 2012*; *Voon et al., 2010*), impairing negative reinforcement (*Cools et al., 2006*; *Frank et al., 2007b*; *Mathar et al., 2017*), or both (*Bódi et al., 2009*; *Maril et al., 2013*; *Palminteri et al., 2009*; *Piray et al., 2014*; *Wittmann et al., 2005*; *Voon et al., 2010*) has been seen multiple times. Likewise, low dopamine conditions have shown either worse positive reinforcement (*Eisenegger et al., 2014*; *Jocham et al., 2011*; *Kobza et al., 2012*), better negative reinforcement (*Bódi et al., 2009*; *Cox et al., 2015*; *Frank et al., 2004*; *Voon et al., 2010*), or both (*Palminteri et al., 2009*; *Piray et al., 2014*).

However, not all PST studies have agreed with these findings. One study found that while PD patients with greater left hemisphere pathology showed dopaminergic medication modulated reward and punishment learning, patients with greater right hemisphere pathology showed no such effects (*Maril et al., 2013*). They also pointed out that left-hemisphere patients are more common, so are likely to be over-represented in study samples unless specifically balanced. We found no effects of laterality of symptoms in our data.

Additionally, the PST was found to have low test-retest reliability when retested 7–8 weeks later (*Baker et al., 2013*); as well as low correlation between performance at different time points, participants initially classed as 'positive learners' or 'negative learners' were labelled differently when retested. The study also failed to find any effects of several dopaminergic genes on RL, in contradiction to previous genetic studies (*Doll et al., 2011*; *Frank and Hutchison, 2009*; *Frank et al., 2007a*). This study questions the idea that dopamine improves positive reinforcement and/or impairs negative reinforcement, and raises doubts about the reliability of the PST.

A recent study has shown that performance on the PST depends on the discriminability of the stimuli used, with the difference in discriminability of stimuli A vs B, and stimuli C vs D affecting whether healthy participants showed choose-A or avoid-B biases (*Schutte et al., 2017*). While the majority of the studies (ours included) using the PST counterbalanced which specific stimuli were A and B (etc.), it is still possible that differences in discriminability between the stimuli within and across experiments may have affected results.

There are many variations of the PST, including using different stimuli (*Waltz et al., 2007*) smiling and frowning faces as feedback (*Aberg et al., 2015*, *2016*; *Gold et al., 2013*; *Jocham et al., 2011*), using money as feedback (*Kunisato et al., 2012*; *Rustemeier et al., 2012*), changing the number of pairs (*Doll et al., 2014*), the probabilities of reward (*Doll et al., 2014*; *Evans and Hampson, 2015*), the number of trials (*Cicero et al., 2014*; *Evans and Hampson, 2015*), and the filtering criterion (*Evans and Hampson, 2015*; *Waltz et al., 2007*). Small changes such as changes to the stimuli used, or changing the delay between action and feedback can have large effects on how this task works (*Foerde and Shohamy, 2011*; *Schutte et al., 2017*). Care should be taken when comparing across such procedures, as we, and others, have shown that small modifications to RL task procedure can have large effects on behaviour.

Some studies are now using simpler RL tasks that require learning to predict the outcome associated with one stimulus shown, rather than picking from two simultaneously shown stimuli (*Bódi et al., 2009*; *Herzallah et al., 2013*; *Mattfeld et al., 2011*; *Simon and Gluck, 2013*; *Tomer et al., 2014*). Tasks including punishments as well as rewards are frequently used, and often have a simpler probabilistic structure with the same probability for pairs used for the reward and punishment pairs (*Eisenegger et al., 2014*; *Naef et al., 2017*; *Palminteri et al., 2009*; *Pessiglione et al., 2006*). These simpler tasks may make it easier for participants to learn, and have

lesser effects of stimulus discriminability. However, discriminability effects have not been tested for in these tasks, and to date only the Weather Prediction Task has published test-retest reliability data (*Aron et al., 2006*). So whether other RL tasks are actually better than the PST or have the same issues remains to be seen. It would be useful if reliability and validation data were included in publications for tasks such as these in future.

It is interesting to consider what the implication of our results are for the theories of basal ganglia function. In particular, how our results can be reconciled with the observations that dopaminergic neurons activate striatal D1 neurons, which are believed to be involved in activating movements, while they inhibit the D2 neurons, which are thought to be involved in movement inhibition (*Kravitz et al., 2010*). It has been recently proposed that these neurons encode not only the expected value of an action in the difference of their activity, but also the variance of the reward in the sum of their activity (*Mikhael and Bogacz, 2016*). In PST, the stimuli with reward probability closer to 50% have a high variance of reward, because on some trials they result in positive and on some trials in negative feedback. In the simulations of the PST these stimuli strongly activated both D1 and D2 neurons (*Mikhael and Bogacz, 2016*). Thus on the simulated novel-pair trials with such stimuli and the stimulus A, the D1 neurons selective for both options were activated, and hence increasing the level of dopamine had little affect the accuracy in choose-A (or even decreased it for some variants of the model). These simulations show that the level of dopamine may little influence the accuracy the PST even if the dopaminergic modulation differentially affects the striatal D1 and D2 neurons.

## Conclusion

Dopamine and PD did not affect expression of positive or negative reinforcement when tested immediately or 24 hr after learning. Dopamine during learning improved the consolidation of RL memories over 24 hr. The original PST had very low accuracy, and the modifications made to the PST had large effects, increasing learning and novel pairs accuracy, and increasing the amount of avoid-B selections participants made. This highlights the effects that can be induced by small changes to these types of tasks. These experiments failed to replicate the previously reported effects of dopamine and PD on RL, suggesting the effect may be weak.

## Materials and methods

### Experiment 1

Ethical approval was obtained from the NHS Research Ethics Committee at Frenchay, Bristol. All participants gave written consent, in accordance with the Declaration of Helsinki.

### Participants

The three experiments reported here all tested PD patients and HC. Demographic data are shown in *Table 1*. PD patients had a diagnosis of idiopathic PD, were stable on their medications for at least 3 months, and were recruited through the general neurology and movement disorders clinic in Frenchay and Southmead Hospitals, Bristol. Patients were on levodopa and/or dopamine agonists, were not taking mono-amine oxidase inhibitors and did not have a Deep Brain Stimulator implanted. When coming OFF medication, patients were withdrawn from standard-release medication a minimum of 15 hr prior to testing, and from prolonged-release medication a minimum of 24 hr prior to testing. For the OFF-OFF condition patients went back ON medication for a short while after the day 1 session, to minimise time spent OFF medication and the risk of neuroleptic malignant syndrome (*Keyser and Rodnitzky, 1991*).

HC were recruited from the ReMemBr Group's healthy volunteer database.

Experiment 1 tested 18 PD patients and 18 HC. PD patients were tested in a within-subjects manner, with all patients tested in all medication conditions.

### Procedure

PD patients were tested over two days, with learning taking place on day 1, and testing on day 2. Patients could be ON or OFF for each of these days, giving four conditions (ON-ON, ON-OFF, OFF-

**Table 1.** The means and SEM for each experiment for PD patients and HC on all measures taken. Within each experiment, one-way ANOVAs were run between PD patients and HC ($\chi^2$ for the genders), and paired t-tests for the comparisons between patients ON and OFF medication for the UPDRS. MMSE scores from experiment 1 were converted to MoCA scores for comparison. *p<0.05, **p<0.01, ***p<0.001.

| Experiment | 1 | | 2 | | 3 | |
|---|---|---|---|---|---|---|
| Measure | PD patients | HC | PD patients | HC | PD patients | HC |
| Number | 18 | 18 | 18 | 20 | 18 | 18 |
| Gender (M/F) | 15/3** | 7/11 | 16/2*** | 5/15 | 11/7 | 11/7 |
| Age | 71.56 (2.06) | 71.19 (2.52) | 67.39 (2.10) | 66.05 (2.05) | 69.11 (1.44) | 71.61 (2.05) |
| Years Education | 13.50 (0.66) | 12.93 (0.89) | 14.83 (0.91) | 13.75 (0.56) | 11.94 (0.52)* | 14.72 (0.65) |
| MoCA | 29.44 (0.12) | 29.63 (0.13) | 28.72 (0.50)* | 26.85 (0.53) | 27.61 (0.54) | 26.78 (0.56) |
| DASS | 21.71 (2.85)* | 12.19 (2.76) | 15.39 (2.54) | 20.05 (5.50) | 29.13 (4.53)** | 10.44 (2.29) |
| Depression | 6.35 (0.81)** | 2.88 (0.87) | 4.94 (1.13) | 5.55 (1.958) | 7.13 (1.71)*** | 2.78 (0.75) |
| Anxiety | 7.88 (1.27)** | 3.25 (0.88) | 5.67 (0.84) | 5.50 (1.80) | 10.33 (1.58)*** | 2.61 (0.69) |
| Stress | 7.47 (1.41) | 6.06 (1.32) | 4.78 (1.09) | 9.00 (2.02) | 11.67 (1.73)** | 5.06 (1.25) |
| BIS | 14.94 (2.36) | 15.31 (3.07) | 53.50 (2.47) | 51.90 (2.35) | 53.56 (2.70) | 51.00 (1.88) |
| LARS | −20.22 (1.36)*** | −27.44 (1.24) | −23.50 (1.71)* | −30.00 (1.58) | −22.44 (2.06)* | −27.06 (1.16) |
| UPDRS ON | 18.67 (2.69) | | 18.78 (2.85) | | 26.50 (2.73) | |
| UPDRS OFF | 24.56 (3.41)*** | | 23.28 (2.97) | | 30.44 (2.52)*** | |
| Years since diagnosis | 4.44 (1.21) | | 4.39 (0.90) | | 5.00 (1.02) | |
| Years since symptoms | 5.18 (1.28) | | 4.78 (0.86) | | 6.44 (1.10) | |
| LDE (mg) | 566.41 (61.18) | | 543.70 (67.36) | | 653.00 (93.96) | |
| # levodopa/ dopamine agonists/both | 12/1/5 | | 10/2/6 | | 10/0/8 | |
| # on XL meds | 3 | | 10 | | 9 | |

MoCA=Montreal Cognitive Assessment, DASS=Depression, Anxiety and Stress Scale, BIS=Barratt Impulsivity Scale, LARS=Lille Apathy Rating Scale, UPDRS=Unified Parkinson's Disease Rating Scale, LDE=Levodopa Dose Equivalence.

ON, OFF-OFF; see *Figure 6*) tested in a randomised, counterbalanced order. All patients completed all four conditions. HC completed one pair of days.

Presentation software (Version 18.0, Neurobehavioural Systems, Inc., RRID:SCR_002521) was used to run all experiments. A modified version of the PST was used, as during piloting it was found that participants could not learn the standard PST. In the modified PST, only 2 rather than 3 pairs were used, with the probabilities of rewards for the cards 80% and 20% and 65% and 35%. Instead of written feedback, smiling and frowning faces were used (see *Figure 7*). On each learning trial participants saw two symbols and had to select one with a button press. If they hadn't responded within two seconds they were shown a 'GO' prompt, to prompt faster responding. There were four versions of the task, with different Hiragana symbols for each one, so that patients saw different symbols in each condition's session. Which symbol in each pair was actually card A and card B was counterbalanced (same for CD and EF) and the versions given in each medication condition were randomised.

After 40 practice trials (different stimuli), and 240 learning trials (three blocks of 80), there was a block of 40 'memory trials' which were the same as the learning trials but without the feedback. This measured the participants' memory on the learning pairs, and is analogous to the delayed memory trials from *Coulthard et al. (2012)*. Another memory block was given 30 min later.

On day 2, 24 hr after learning, a third memory block was given, and then the novel pairs block. This consisted of all the possible pair combinations of the cards (AB, AC, AD, BC, BD, CD) shown without feedback, as in *Frank et al. (2004)*. Each was shown 15 times for a total of 90 trials. From this, the percentage of times that card A was shown and chosen (choose-A) was taken as a measure

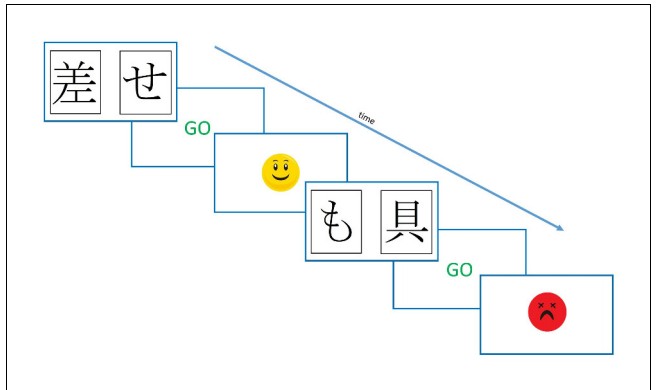

**Figure 7.** The modified Probabilistic Selection Task procedure. Participants saw two symbols on the screen, and selected one with a button press. If no response was made within 2 s, they were shown a 'GO' prompt. Feedback was determined probabilistically, was shown for 2 s, and was either the smiling or frowning face.

of positive reinforcement, and the percentage of times that card B was shown and avoided (avoid-B) was taken as a measure of negative reinforcement.

In addition to the PST, several other tasks were administered including the HVLT-R, UPDRS, MMSE, DASS, LARS, BIS and SMHSQ.

## Experiment 2

As experiment 1 failed to show predicted effects of medication or disease state, further experiments were run. As we had made some modification to the PST based on our pilot data, we designed experiment 2 to test whether performance using our modified PST was as predicted from previous work (e.g. *Frank et al., 2004*) with no delay between learning and testing.

### Participants

Eighteen PD patients and 20 HC were tested in experiment 2. Demographic data are shown in *Table 1*. Six of the PD patients were previously tested in experiment 1, however excluding these from the analysis did not change the direction of effects, and did not materially affect statistical outputs, so these participants were included in the analysis.

Ten PD patients were given the MMSE while the others, and all HC, were given the MoCA. MMSE scores were converted to MoCA scores using the PD conversion from *Waltz et al. (2007)*.

### Procedure

The modified PST described above was used again, but this time without any memory blocks, and without any delay between the learning and novel pairs blocks. Therefore, participants completed 40 practice trials (different stimuli), then three blocks of 80 learning trials (240 total), followed immediately by the block of 90 novel pairs trials.

As learning and testing was on the same day, PD patients were ON for both, or OFF for both, meaning there were only two testing sessions, and only two conditions for patients. All patients completed both conditions. Again, condition order and task stimuli were randomised and counterbalanced. HC were tested once.

## Experiment 3

### Participants

Eighteen PD patients and 18 HC were tested, demographic data are shown in *Table 1*. None of these participants had taken part in experiments 1 or 2. PD patients were tested once ON and once OFF (randomised order) and HC were tested once.

## Procedure

The original PST (*Figure 1*) was run, with three pairs (80–20%, 70–30%, 60–30%), which gave written 'Correct' or 'Incorrect' feedback. Maximum trial duration was 4000 ms, after which if no response was made, 'No response detected' was printed in red and the next trial began. The feedback was shown for 2000 ms.

Forty practice trials (different stimuli) were given, followed by the learning trials. The learning trials ran in blocks of 60 trials (20 of each pair), and participants had to pass accuracy thresholds in order to exit the training. Within one block, participants had to score above 65% accuracy on the 80–20% pair, 60% accuracy on the 70–30% pair and 50% accuracy on the 60–40% pair. If they did not reach these thresholds by the end of the 7th block, they exited the training anyway (in the original study there were only 6 blocks).

The novel pairs test still had all possible combinations (15), although now each was presented only six times to give a total of 90 trials.

Task stimuli were randomised and counterbalanced.

## Data analysis

In all trials, selections of the cards with the highest probability of reward were taken as the 'optimal' choice, regardless of what feedback they produced on that specific trial. In the novel pairs block, the learning pairs (AB and CD) were excluded from the analysis as in previous studies (*Frank et al., 2004*). Between-subjects ANOVAs were used to compare PD patients to HC, and paired sample t-tests to compare the PD medication conditions (with Bonferroni corrections for multiple comparisons). Cohen's *d* and $\eta_p^2$ effect sizes are given for significant results from t-tests and ANOVAs, respectively.

Additional analyses were conducted after data filtering; if a participant scored 50% or lower on the AB choice in the novel pairs task, they were assumed to have not learnt the task properly and that data were excluded. Each condition was checked separately, so one medication condition's data could be excluded while all other remain. This filtering was only applied to the novel pairs data and analysis.

Some data were missing from the analysis, due to experimenter error or computer errors. Two final learning blocks, two 30 min memory blocks and one novel pairs block were missing from experiment 1.

All error bars in the figures are standard error of the mean (SEM). MATLAB (RRID:SCR_001622) was used for data processing (code available at https://github.com/johnPGrogan/Effects-of-dopamine-on-RL-consolidation-in-PD/releases/tag/v1.0; *Grogan, 2017*), and SPSS (RRID:SCR_002865) for statistical tests. A copy of the code is available at https://github.com/elifesciences-publications/Effects-of-dopamine-on-RL-consolidation-in-PD.

## Data availability

We did not obtain consent from participants to share individual data from this study, thus only summary statistics are presented in the figures, tables and text. Individual data are not provided in the source data files, although the summary statistics are.

## Acknowledgements

The authors thank Michael Frank for comments on the manuscript, BRACE charity for the use of their building, and all the patients and participants who took part in the research.

## Additional information

### Funding

| Funder | Grant reference number | Author |
| --- | --- | --- |
| Wellcome | PhD Studentshipt SJ1102 | John P Grogan |
| BRACE | Project grant | John P Grogan<br>Elizabeth J Coulthard |

| Medical Research Council | MC UU 12024/5 | Rafal Bogacz |

The funders had no role in study design, data collection and interpretation, or the decision to submit the work for publication.

### Author contributions

JPG, Conceptualization, Data curation, Formal analysis, Validation, Investigation, Visualization, Methodology, Writing—original draft, Project administration, Writing—review and editing; DT, Conceptualization, Investigation, Methodology, Project administration, Writing—review and editing; LS, BEK, Investigation, Writing—review and editing; RB, Conceptualization, Supervision, Methodology, Writing—review and editing; AW, Conceptualization, Resources, Writing—review and editing; EJC, Conceptualization, Resources, Supervision, Funding acquisition, Methodology, Project administration, Writing—review and editing

### Author ORCIDs

John P Grogan, http://orcid.org/0000-0002-0463-8904

### Ethics

Human subjects: Ethical approval was obtained from the NHS Research Ethics Committee at Frenchay, Bristol (09/H0107/18). All participants gave written consent, in accordance with the Declaration of Helsinki.

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

## Appendix 1

### Win stay lose shift analysis

We analysed performance during the learning trials using the 'win-stay lose-shift' metric. On trials with positive feedback (i.e. a win), we counted how often the participant chose that card the next time it was shown (a stay) or not (a shift), and likewise how often they avoided a card after negative feedback (i.e. shifting after a loss). The number of each behaviour was calculated for each symbol separately, then summed and divided by the number of wins (and losses) to give the percentage of win-stay (and lose-shift) for each participants' condition.

*Appendix 1—figure 1* shows the mean percentages of win-stay and lose-shift during the learning trials for each experiment. A between-subject three-way ANOVA with trial type (win-stay or lose-shift), group (PD ON, PD OFF or HC) and experiment (1, 2 or 3) as factors. PD patients in experiment 1 were grouped by day 1 medication state. There was a significant effect of trial type (F (1, 382)=866.859, p=$2.6881\times10^{-100}$, $\eta_p^2 = 0.694$), with more win-stay than lose-shift. There were no effects of group (F (2, 382)=1.096, p=335, $\eta_p^2 = 0.006$) or experiment number (F (2, 382)=2.235, p=0.108, $\eta_p^2 = 0.012$). The interaction of trial type and experiment number was significant (F (2, 3382)=20.870, p=$2.5014\times10^{-9}$, $\eta_p^2 = 0.099$) suggesting that the different experiments had different patterns of win-stay and lose-shift. All other interactions were not significant (p>0.2).

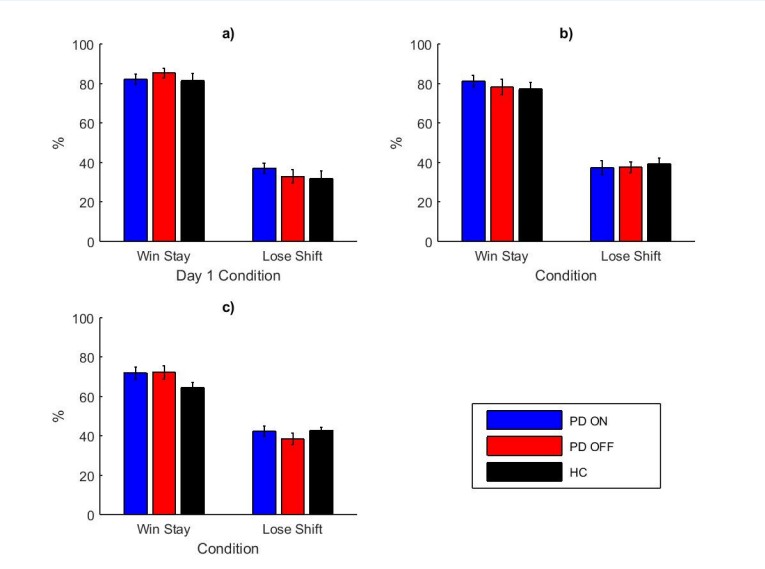

**Appendix 1—figure 1.** The mean percentages of win-stay and lose-shift during the learning trials for experiments 1 (**a**), 2 (**b**) and 3 (**c**), each split by day 1 condition. **Appendix 1—figure 1— source data 1** shows the summary statistics.

**Appendix 1—figure 1—source data 1.** Summary statistics for the percentage of win-stay and lose-shift behaviours during learning trials for each experiment.

Separate two-way ANOVAs on win-stay and lose-shift (with group and experiment number as factors) showed that win-stay was significantly lower in experiment 3 than experiment 1 (Bonferroni-corrected multiple comparisons: p=$6.1235\times10^{-8}$) and experiment 2 (p=0.001), while experiments 1 and 2 did not differ (p=0.219). Lose-shift was significantly higher in

experiment 3 than experiment 1 (p=0.004) but not experiment 2 (p=0.595), and experiments 1 and 2 were not significantly different (p=0.192). Therefore experiment 3 had less win-stay and more lose-shift behaviour than the other experiments, likely due to the differences between the original PST and modified PST (see main text for details).

As mentioned above, *Appendix 1—figure 1* shows that in each experiment, the participants were more often staying after win than shifting after loss. This behaviour may seem surprising given that the participants exhibited greater avoid-B than choose-A behaviour during the novel pairs tests (in experiments 1 and 2, see main text for details). A possible explanation for the low lose-shift rate is that it resulted from reduced switching away from option A when it received negative feedback (on 20% of trials). Namely, the participants might have realised that option A produced correct feedback on average but the feedback was stochastic, so the negative feedback after selecting option A was just noise and should be ignored.

## Appendix 2

## Computational modelling

We fit a variety of computational reinforcement learning models to the participants' learning data from experiments 1–3.

The basic model was the Q-learning model (**Sutton and Barto, 1998**), which assumes that for each stimulus $i$ the participants estimate the expected reward connected with choosing this stimulus, denoted $Q(i)$. This is initialised as 0.5 and is updated after the stimulus is chosen and the reward received for trial $t$:

$$Q_{t+1}(i) = Q_t(i) + \alpha\delta$$

where $\alpha$ is the learning rate parameter ($0 \leq \alpha \leq 1$) and $\delta$ is the reward prediction error, defined as the difference between reward received ($r = 1$ or $0$) and the reward expected:

$$\delta = r - Q_t(i) \tag{i}$$

The model assumes that the probability of choosing stimulus $i$ on trial $t$ depends on the estimated values of the stimuli in the following way:

$$P(i) = \frac{e^{\beta Q_t(i)}}{\sum_{i=1}^{n} e^{\beta Q_t(i)}}$$

where $\beta$ is the inverse temperature of the softmax equation, and controls how deterministic the selection is. Low $\beta$ values lead to random choices, while high $\beta$ values lead to stimuli with higher estimated values being chosen.

We also included a dual learning rate model which has separate learning rates for positive and negative prediction errors:

$$\begin{cases} Q_{t+1}(i) = Q_t(i) + \alpha_+\delta & \delta > 0 \\ Q_{t+1}(i) = Q_t(i) + \alpha_-\delta & \delta < 0 \end{cases}$$

This allows the model to capture different rates of learning from positive and negative reinforcement.

For each patient, we fit all the model to the data from all four conditions from experiment 1, using the same parameters for all conditions. We also fit the models with different learning rate parameters for day 1 medication state. This gave us four models in total:

1. Single ($\alpha$) plus $\beta$ (2 parameters total)
2. Dual learning rates ($\alpha_+$ and $\alpha_-$) plus $\beta$ (3 parameters in total)
3. Single learning rate for each day 1 condition ($\alpha_{ON}$ and $\alpha_{OFF}$) plus $\beta$ (3 parameters in total)
4. Dual learning rates for each day 1 condition ($\alpha_{ON+}$, $\alpha_{OFF+}$, $\alpha_{ON-}$ and $\alpha_{OFF-}$) plus $\beta$ (5 parameters in total)

The parameters of each model were found for which the participants' choices were most likely (**Daw, 2011**). The negative log likelihood was minimised using MATLAB's fminsearch function and the initial parameters were generated randomly from uniform distribution on range [0, 1]. For each model, this fitting procedure was repeated 20 times using different sets of randomly generated initial parameters, to avoid local minima. Bayesian Information Criteria (BIC) (**Schwarz, 1978**) were used to compare the fits of the models. For each PD patient, the models were fit to the learning data from all their medication conditions

together (4 conditions in experiment 1, 2 conditions in experiments 2 and 3). HC only had one session, so the models with separate parameters for different medication states could not be fit to them.

The dual learning rate Q-learning rate model (model #2) was the best fitting model (lowest BIC) for each experiment (see *Appendix 2—table 1* for mean BIC and parameter values). This model had no separate parameters for the different medication states during learning. The positive learning rates were significantly higher than the negative learning rates for this model for all three experiments (p<.005), reflecting the higher rate of win-stay than lose-shift behaviour discussed above.

**Appendix 2—table 1.** The mean BIC and model parameter values for each model for each experiment. The bolded lines show the models with the smallest BIC, which was model number 2, the dual-learning rate model without separate parameters for day 1 conditions.

| Experiment | Model | BIC | on | off | α+ON | α+OFF | α-ON | α-OFF | β |
|---|---|---|---|---|---|---|---|---|---|
| 1 | 1 | 996.243 | 0.1006 | | | | | | 60.8972 |
| | **2** | **885.9434** | | | **0.1024** | | **0.0041** | | **9.678** |
| | 3 | 992.3985 | 0.0808 | 0.0988 | | | | | 36.866 |
| | 4 | 891.5522 | | | 0.1184 | 0.1593 | 0.0184 | 0.0097 | 7.3885 |
| 2 | 1 | 520.6029 | 0.1131 | | | | | | 231.6615 |
| | **2** | **485.0105** | | | **0.1922** | | **0.0319** | | **21.278** |
| | 3 | 510.5098 | 0.1248 | 0.0616 | | | | | 116.4691 |
| | 4 | 487.6825 | | | 0.3144 | 0.1825 | 0.0438 | 0.1273 | 7.2812 |
| 3 | 1 | 807.0322 | 0.132 | | | | | | 101.1408 |
| | **2** | **737.4145** | | | **0.2069** | | **0.0253** | | **5.9512** |
| | 3 | 802.9193 | 0.1059 | 0.0624 | | | | | 163.4829 |
| | 4 | 746.7926 | | | 0.2918 | 0.1846 | 0.0521 | 0.0555 | 4.104 |

Models 1 and 2 were also fit to the HC data from each experiment (models 3 and 4 could not be fit as there were no medication conditions). In all experiments, model 2 was the best fit with lower BIC values (experiment 1: model 1 = 242.0490, model 2 = 220.4385; experiment 2: model 1 = 256.0216, model 2 = 244.3703; experiment 3: model 1 = 265.3291, model 2 = 256.5143). Over all three experiments model 2 had a lower BIC (model 1 = 254.8948, model 2 = 241.1308). The positive learning rates were significantly larger than the negative learning rates in model 2 for all three experiments (p<0.0001).

The BIC values were much lower for the HC fits than the PD fits, because HC fits only included one session, so the total log likelihood of the data was a sum of fewer terms corresponding to likelihoods of individual trials. To compare directly, we fit the single and dual learning rate Q-learning models to each medication condition separately for each PD patient. This showed no significant difference in BIC values between patients and controls for either model in experiment 1 (p>0.9) or 2 (p>0.8). In experiment 3, the dual learning rate model gave borderline significantly lower BIC values for PD patients than HC (p=0.0578), although the single learning rate model did not (p=0.1241). This suggests that overall the models fit equally well to PD patients and HC.

