## [Decision Letter]

Thank you for submitting your article "Effects of dopamine on reinforcement learning and consolidation in Parkinson's disease" for consideration by *eLife*. Your article has been favorably evaluated by Timothy Behrens (Senior Editor) and three reviewers, one of whom is a member of our Board of Reviewing Editors. The following individual involved in review of your submission has agreed to reveal his identity: Travis Baker (Reviewer #3).

The reviewers have discussed the reviews with one another and the Reviewing Editor has drafted this decision to help you prepare a revised submission.

Summary:

This study follows a number of previous studies that explored the relationship between dopaminergic medication and learning, including examining in more detail the main effect described by Frank and colleagues in 2004. As the authors note, it is not clear from that original study whether the effects of dopamine were related to learning during a probabilistic selection task (PST), or rather to the consolidation and retrieval of the learned values. Here they use three separate experiments to test how medication influences both memory and learning from positive and negative feedback. The most striking result is a failure to replicate the primary findings from Frank and colleagues. They also present a novel finding showing that patients on mediation during learning had increases in memory accuracy after waiting 24 hours for testing, suggesting a role for dopamine in memory consolidation but not necessarily on reinforcement learning.

The reviewers agree that this is a highly worthwhile and well-executed study, and the manuscript is well written. The authors use sound methods but fail to reproduce a highly cited study. As such, this study has the potential to help move the field forward by better understanding the exact conditions in which dopamine affects learning, memory, and decision-making behavior. That said, the reviewers also agree that there are several major issues that must be addressed, detailed below.

Essential revisions:

1) Given in particular the lack of learning by participants in Experiment 3, it would be useful to have a more thorough analysis and discussion of the differences in performance between the testing and training phase for all of the experiments. It may be worthwhile to use certain learning models (e.g., Q-learning) to characterize learning behavior under the various conditions to better understand the lack of overall learning in Experiment 3 and more generally relationships between performance in the training and testing phases of the experiments.

2) Given the positive findings about memory consolidation, it would also be useful to include a more thorough analysis and discussion of the relationship between task performance and working memory. Are there behavioral patterns in the train/test and on/off conditions that can be related more directly to working memory function (e.g., win-stay/lose-shift)? How does their interpretation relate to other findings that relate dopamine signaling to memory formation? These kinds of results might be interpreted in the context of findings showing projections of midbrain dopamine neurons to the hippocampus and to the surrounding MTL cortices (Samson et al., 1990; Gasbarri et al., 1994) and may contribute to successful binding between experiences separated by time (Cohen and Eichenbaum, 1993; Shohamy and Wagner, 2008). Such binding, mediated by tonic dopamine signals (Niv et al., 2007), begins before the experiences and continues into a temporal window of hours or days (Shohamy and Adcock, 2010). Foerde and Shohamy (2011), in an fMRI study of healthy young adults performing a probabilistic learning task, demonstrated the recruitment of the striatum during learning with immediate feedback, and increased activation of the hippocampus with delayed feedback. Data from the same authors showed that individuals with Parkinson's disease, whose striatum is known to be degraded, were impaired in learning from immediate but not delayed feedback (Foerde and Shohamy, 2011). Conversely, individuals with MTL damage exhibited impaired learning with delayed but not immediate feedback (Foerde et al., 2013).

3) In general, the paper could benefit from a more thorough vetting of the statistical analyses and claims, including:

a) specifying error bars in all of the figures;

b) appropriately interpreting "borderline" significance for the effect of day 1 medication state (also, do non-parametric tests yield the same results?);

c) clarifying (and possibly re-interpreting) the claim that "both day 1 ON conditions (blue bars) increased in memory scores," which is also repeated in the Discussion but seems to run counter to the OnOn data presented in Figure 2 (which does not seem to differ significantly from zero, given the error bars shown);

d) clarifying statements in Results ("The pattern of day 1 ON patients showing more avoid-B than day 1 OFF patients is in the opposite direction to predictions from previous work") and Discussion ("day 1 ON conditions having the highest amount of avoid-B selections") that appear to be contracted by the actual findings ("There were no significant effects of day 1 or day 2 medication state, or any interactions (p >.28). This suggests that […] medication on day 1 or day 2 had no effect.").

4) To effectively compare the results with previous findings, the claim that "our samples were very closely matched in age, gender and disease severity to the PD patients tested ON medication in Frank et al. (2004)" needs to be fleshed out more. What, exactly, were the comparisons? How well did they match, particularly disease severity?

5) It would be useful to include a more thorough discussion of the limitations of the PST task, including what future directions might help either validate the task as an effective way to study mechanisms of reinforcement learning, or point the way to new, more effective task designs.

[Editors' note: further revisions were requested prior to acceptance, as described below.]

Thank you for resubmitting your work entitled "Effects of dopamine on reinforcement learning and consolidation in Parkinson's disease" for further consideration at *eLife*. Your revised article has been favorably evaluated by Timothy Behrens (Senior Editor) and a Reviewing Editor.

The manuscript has been improved greatly but there is one remaining issue that needs to be addressed before acceptance, as outlined below:

The Q-learning fits are a welcome addition and do a nice job of showing that, among the models tested, the one that uses learning rates separated for positive and negative reinforcement but not ON versus OFF medication best fit the data. However, the fits also suggest that the model fits the HC data much better than the patient data (substantially lower BIC values). This suggests that HC and PD patients might be using different strategies – a point that should be noted, and its implications discussed.

---

## [Author Response]

*Essential revisions:*

*1) Given in particular the lack of learning by participants in Experiment 3, it would be useful to have a more thorough analysis and discussion of the differences in performance between the testing and training phase for all of the experiments. It may be worthwhile to use certain learning models (e.g., Q-learning) to characterize learning behavior under the various conditions to better understand the lack of overall learning in Experiment 3 and more generally relationships between performance in the training and testing phases of the experiments.*

We have fitted several RL models to the data, and found the best fitting one to be a Q-learning model with two learning rates that did not depend on medication state. We have stated this in the manuscript (“Experiment 1” subsection “Learning”, last paragraph) and provided fuller details in the Appendix 2.

We have also looked for correlations between final learning block accuracy and overall accuracy on the novel pairs tests. As would be expected, there are medium/strong positive correlations for each experiment, as better learning leads to better test accuracy. This is stated in each experiment’s ‘Novel Pairs’ section.

*2) Given the positive findings about memory consolidation, it would also be useful to include a more thorough analysis and discussion of the relationship between task performance and working memory. Are there behavioral patterns in the train/test and on/off conditions that can be related more directly to working memory function (e.g., win-stay/lose-shift)?*

We have examined win-stay lose-shift behaviour in the learning trials of each experiment. These did not show any significant differences between PD conditions, although Experiment 3 did differ from Experiments 1 and 2 in this regard. We have stated in the Experiment 1 results (subsection “Learning”, second paragraph) that we found no significant differences between groups in win-stay lose-shift behaviour, and have given the full details in Appendix 1.

*How does their interpretation relate to other findings that relate dopamine signaling to memory formation? These kinds of results might be interpreted in the context of findings showing projections of midbrain dopamine neurons to the hippocampus and to the surrounding MTL cortices (Samson et al., 1990; Gasbarri et al., 1994) and may contribute to successful binding between experiences separated by time (Cohen and Eichenbaum, 1993; Shohamy and Wagner, 2008). Such binding, mediated by tonic dopamine signals (Niv et al., 2007), begins before the experiences and continues into a temporal window of hours or days (Shohamy and Adcock, 2010). Foerde and Shohamy (2011), in an fMRI study of healthy young adults performing a probabilistic learning task, demonstrated the recruitment of the striatum during learning with immediate feedback, and increased activation of the hippocampus with delayed feedback. Data from the same authors showed that individuals with Parkinson's disease, whose striatum is known to be degraded, were impaired in learning from immediate but not delayed feedback (Foerde and Shohamy, 2011). Conversely, individuals with MTL damage exhibited impaired learning with delayed but not immediate feedback (Foerde et al., 2013).*

We are grateful to the reviewers for pointing out the literature, and have made significant changes to incorporate it. We have expanded on the interpretation of the memory consolidation result in the Discussion (subsection “Effects of dopamine on consolidation”, third paragraph), linking it to the several of the references given here, as well as animal literature showing beneficial consolidation effects of dopaminergic drugs infused after learning. We have also mentioned the feedback timing study (Foerde & Shohamy, 2011) in here, to link this task to the striatum, and again later in the paragraphs about limitations of the PST.

*3) In general, the paper could benefit from a more thorough vetting of the statistical analyses and claims, including:*

*a) specifying error bars in all of the figures;*

We have specified SEM bars in each legend, and in the “Data analysis” subsection (last paragraph).

*b) appropriately interpreting "borderline" significance for the effect of day 1 medication state (also, do non-parametric tests yield the same results?);*

We have removed the interpretation of borderline and direction-only effects from the Results section, and included non-parametric Wilcoxon’s tests for the memory data (Experiment 1 subsection “Memory”, last paragraph), which gave very similar results.

*c) clarifying (and possibly re-interpreting) the claim that "both day 1 ON conditions (blue bars) increased in memory scores," which is also repeated in the Discussion but seems to run counter to the OnOn data presented in Figure 2 (which does not seem to differ significantly from zero, given the error bars shown);*

We have re-written this claim (Experiment 1 subsection “Memory”, second paragraph) to state that the means of the groups show a slight increase for day 1 ON groups, while HC and day 1 OFF show mean decreases, and point out that the SEM bars overlap for some of these groups. The statistics are then presented to show which differences are significant. We have also rewritten statements in the Abstract and Discussion (subsection “Effects of dopamine on consolidation”, first paragraph) to reflect that day 1 ON didn’t increase in memory, but rather did not show the decrease day 1 OFF conditions showed.

*d) clarifying statements in Results ("The pattern of day 1 ON patients showing more avoid-B than day 1 OFF patients is in the opposite direction to predictions from previous work") and Discussion ("day 1 ON conditions having the highest amount of avoid-B selections") that appear to be contracted by the actual findings ("There were no significant effects of day 1 or day 2 medication state, or any interactions (p >.28). This suggests that […] medication on day 1 or day 2 had no effect.").*

We have removed this statement from the Results, and clarified the one in the Discussion (subsection “Effects of dopamine on learning from positive and negative feedback”, first paragraph) to make it clear that the direction of effects is in the opposite to what was predicted, rather than it being the same direction but simply a small difference that may not have been detected due to lack of power from small sample size.

*4) To effectively compare the results with previous findings, the claim that "our samples were very closely matched in age, gender and disease severity to the PD patients tested ON medication in Frank et al. (2004)" needs to be fleshed out more. What, exactly, were the comparisons? How well did they match, particularly disease severity?*

We have run t-tests on the sample characteristics provided in the supplementary information of Frank et al. (2004). These showed that our Experiment 3 PD sample was not significantly different in terms of age or Hoehn & Yahr staging, and χ^2^ tests showed no difference in gender distribution. There was a significant difference in the years of education, with our participants having fewer, which could have contributed to their poor learning, however our Experiment 1 sample also had fewer years of education than the Frank et al. 2004 sample and were able to learn the modified PST fine. This is stated in the manuscript (subsection “Effects of dopamine on learning from positive and negative feedback”, fifth paragraph).

*5) It would be useful to include a more thorough discussion of the limitations of the PST task, including what future directions might help either validate the task as an effective way to study mechanisms of reinforcement learning, or point the way to new, more effective task designs.*

We have expanded this part of the Discussion (subsection “Effects of dopamine on learning from positive and negative feedback”, ninth paragraph), mentioning the recent Schutte et al. (2017) PLOS ONE paper that shows the discriminability of the specific stimuli used in the task has a large effect on the choose-A/avoid-B behaviour. We also mention the Foerde & Shohamy (2011) paper, suggested in point 2 above, showing that feedback timing in a similar probabilistic learning task (tenth paragraph of the aforementioned subsection).

We have mentioned some of the other RL tasks that are being used frequently, such as those with simpler probabilistic structures, or gains and losses rather than written feedback. We suggest that, in future, these types of tasks are presented with some form of validation or reliability analysis, as this is currently very rare in the RL task literature (eleventh paragraph of the aforementioned subsection).

[Editors' note: further revisions were requested prior to acceptance, as described below.]

*The manuscript has been improved greatly but there is one remaining issue that needs to be addressed before acceptance, as outlined below:*

*The Q-learning fits are a welcome addition and do a nice job of showing that, among the models tested, the one that uses learning rates separated for positive and negative reinforcement but not ON versus OFF medication best fit the data. However, the fits also suggest that the model fits the HC data much better than the patient data (substantially lower BIC values). This suggests that HC and PD patients might be using different strategies – a point that should be noted, and its implications discussed.*

The HC fits have lower BICs because they are fit only to a single session’s learning trials, while the fits to the PD data contain the learning trials from all the session they have completed (4 in Experiment 1, 2 in Experiments 2 and 3). This is to allow some parameters to be shared across medication conditions during the fitting. Including more sessions increases the negative log likelihood, as well as increasing the penalty for number of trials included in the BIC.

If we fit each condition for the PD patients separately, then the BIC values are very similar to the HC single-session fittings. Experiments 1 and 2 do not have significantly different BIC values for PD patients or HC in this case (p >.8), while Experiment 3 has borderline significantly lower BIC values for PD patients for the dual learning rate Q-learning model (p =.0578), and no significant difference for the single learning rate model (p =.1241).

We have clarified that the PD model fitting has multiple sessions in each fit (“Appendix 2 Computational Modelling”, sixth paragraph), and therefore more trials which will increase the BIC. We have also included a paragraph (last paragraph of the aforementioned Appendix) explaining the difference in values presented in the paper, and detailing the single-session PD fittings and statistical comparisons with HC fittings mentioned in the paragraph above this one.

The BICs for models 1 and 2 were mixed up for HCs from experiment 1, so we have also corrected this, and report now that for HCs from all experiments BICs were lower for model 2 than model 1 (eighth paragraph of the aforementioned Appendix).